# ENHANCING HUMAN-AI COLLABORATION THROUGH LOGIC-GUIDED REASONING

**Chengzhi Cao[1,2\*], Yinghao Fu[1\*], Sheng Xu[1], Ruimao Zhang[1], Shuang Li[1†]**
[1]The Chinese University of Hong Kong (Shenzhen)
[2]University of Science and Technology of China
`chengzhicao@mail.ustc.edu.cn, yinghaofu@link.cuhk.edu.cn,`
`shengxu1@link.cuhk.edu.cn, zhangruimao@cuhk.edu.cn`
`lishuang@cuhk.edu.cn`

## ABSTRACT

We present a systematic framework designed to enhance human-robot perception and collaboration through the integration of logical rules and Theory of Mind (ToM). Logical rules provide interpretable predictions and generalize well across diverse tasks, making them valuable for learning and decision-making. Leveraging the ToM for understanding others' mental states, our approach facilitates effective collaboration. In this paper, we employ logic rules derived from observational data to infer human goals and guide human-like agents. These rules are treated as latent variables, and a rule encoder is trained alongside a multi-agent system in the robot's mind. We assess the posterior distribution of latent rules using learned embeddings, representing entities and relations. Confidence scores for each rule indicate their consistency with observed data. Then, we employ a hierarchical reinforcement learning model with ToM to plan robot actions for assisting humans. Extensive experiments validate each component of our framework, and results on multiple benchmarks demonstrate that our model outperforms the majority of existing approaches.

## 1 INTRODUCTION

Humans are natural mind-readers. The capacity to deduce what others are thinking or wanting from brief nonverbal interactions, called Theory of Mind (ToM), is crucial to our social lives (Mao et al., 2023; Fuchs et al., 2023). We know not only how to take actions to interact with the world but, moreover, how to *purposefully* influence the thoughts of other humans (collaborators or opponents) to achieve our goal (Chandra et al., 2020). This intellectual ability enables humans to work smoothly and strategically with other humans, which distinguishes humans from other lower-level animals. An intriguing question arises:

*Could we endow AI agents with this ability so that the AI agents can be more intellectually human-like?*

Particularly in a human-AI collaborative task setting, if AI agents can adeptly reason about the unobservable mental states of humans, they could engage more effectively in cooperative tasks, facilitating seamless human-AI interactions (Rabinowitz et al., 2018; Grassiotto & Costa, 2021). For example, a self-driving car equipped with the ToM can better communicate with other human drivers nonverbally. Faced with sophisticated or ambiguous traffic scenarios, ToM enables self-driving cars to reason about the mental states, specifically the beliefs, of human drivers sharing the road (Patel & Chernova, 2022). For another instance, suppose one day we aim to develop an AI agent to aid and take care of elder people, it requires understanding human nonverbal communication, such as hand gestures, or eye contact, to understand the intentions or desires of humans. An agent equipped with ToM is endowed with the ability to interpret these cues more effectively. To achieve this goal, the robots must develop two key skills: 1) comprehend human behavior (called social perception);

---

*Equal Contribution
†Corresponding author

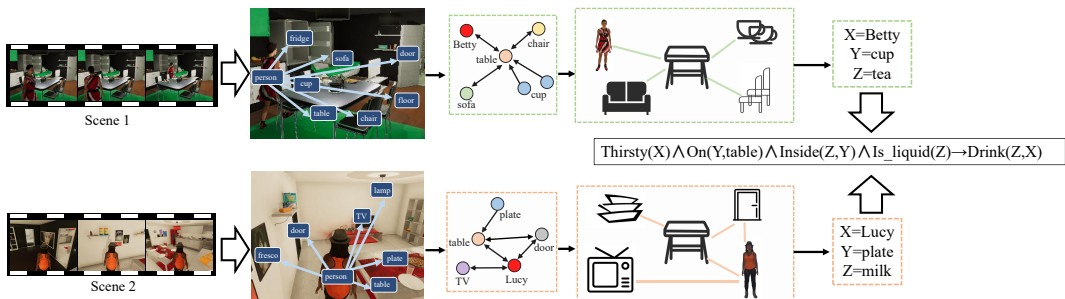

Figure 1: Using logical rules (shown in the black box) for human action perception.

2) reason critically about the environment and organize its actions to work with humans (Puig et al., 2020). Researchers typically refer to the two agents engaged in behavior modeling as "actor" and "observer". The actor behaves based on his own perception of the world, then the observer watches the actor and forms an understanding of the mental state held by the actor and/or the resulting actions that the actor intends to take. In the simplest case, the actor behaves deterministically, and the observer has full knowledge of the world external to the actor (Wei et al., 2018). In more complex cases, observers can have partial knowledge, and agents can be both actors and observers at the same time, simultaneously modeling each other.

Complex social movements often have underlying logic that is clear and generalizable. Logic rules serve as high-level representations for reasoning, and guiding actions in specific scenarios (Duan et al., 2022; Yang et al., 2017). These interpretable rules not only aid in understanding inference outcomes but also enhance robustness in diverse situations. The burgeoning field of neurosymbolic AI exemplifies this trend, merging neural networks with Symbolic AI to foster robust and adaptable models. It maintains the validity of logic rules across various contexts without retraining, contrasting with methods reliant on specific entity embeddings (Kautz, 2022; d'Avila Garcez & Lamb, 2020). Figure 1 illustrates an example of rules. If new facts about more scenes or locations are added to the knowledge base, the rule about "Drink(person,water)" will still be usefully accurate without retraining. The same might not be true for methods that learn embeddings for specific knowledge base entities.

In this paper, we leverage probabilistic logical reasoning to generate generalized knowledge about an agent's goal in new environments, enhancing social perception and human-AI collaboration. This approach, grounded in probabilistic logic, is adept at handling uncertainties and modeling complex data. Our methodology intertwines perception and collaboration learning, where perception informs logic rule learning, and logical reasoning guides collaborative efforts. This reciprocal enhancement offers two key advantages: (*i*) Rule-based Probabilistic Social Perception: Our model treats each hidden rule as latent variables based on probabilistic assessments. These rules capture both object relations and event sequence constraints. (*ii*) Dynamic Collaboration: Our human-like agent adapts its subgoals and beliefs in real-time, guided by the latest observations and logical rules. It formulates and executes concurrent plans alongside other agents, ensuring robust task performance in partially observable environments and generating human-like behaviors.

## 2 RELATED WORK

**Behavior understanding.** The task of recognizing and interpreting human activities faces significant challenges due to the variability in execution styles and the complexity of visual stimuli. Previous research primarily centered on visual behavior modeling (Chen et al., 2021) and trajectory prediction (Cao et al., 2023; Gupta et al., 2018; Alahi et al., 2016; Chandra et al., 2020), with some studies focusing on group activities and plans (Ibrahim et al., 2016; Shu et al., 2015; Choi & Savarese, 2013). Our approach diverges from these works by focusing on activity understanding that infers mental states (e.g., intentions, desires) from behaviors, aligning closely with computational Theory of Mind (Jara-Ettinger, 2019; Rabinowitz et al., 2018; Nguyen et al., 2022). We aim to synergize behavior understanding with logical reasoning, enhancing the machine's comprehension of human actions through a unified, interactive framework.

**Logical reasoning.** The integration of logical reasoning with deep learning architectures has garnered considerable interest (Kautz, 2022; d'Avila Garcez & Lamb, 2020). Most existing algorithms develop differentiable modules based on known relationships for deep networks to learn (Zhou, 2019; Mao et al., 2023; Evans & Grefenstette, 2018). Innovations in this field include learning numerical rules from knowledge graphs (Wang et al., 2019b), differentiable maximum satisfiability solvers (Wang et al., 2019a), and neural-symbolic reasoning models (Tang et al., 2022). However, these models often simplify relations and rely on basic triplet-based embeddings. Our approach advances this field by embedding more intricate first-order logical rules and jointly modeling entities and relations.

**Human-robot interaction.** Historically, human-robot interaction studies have been confined to controlled lab settings (Goodrich & Schultz, 2007; Dautenhahn, 2007). Notable developments include frameworks for learning from joint-action demonstrations (Nikolaidis et al., 2015) and modulating robot behavior based on user actions (Rozo et al., 2016). Other approaches, like natural language interaction (Gervasi et al., 2020; Ogenyi et al., 2021), focus on creating user models within a Partially Observable Markov Decision Process (POMDP) framework. Our research departs from these methods by utilizing logic rules to infer human goals and efficiently associate new users in diverse environments.

## 3 MODEL

### 3.1 PROBLEM STATEMENT

In this system, each individual represents an integrated agent who formulates their actions based on their individual goals, their beliefs about the possible actions of another agent, and a complete observation of the environment. The robot, on the other hand, has access to a single demonstration of a human performing the same task in the past, as well as a complete observation of the environment. Consequently, at each step, the robot receives observations from the system and issues action commands to control the avatar within the environment. Formally, each task is defined by the person's goal $g$ (a set of goal predicates), a demonstration of Alice taking actions to achieve that goal $D = \{\mathbf{s}^t, \mathbf{a}^t\}_{t=1}^T$ (a sequence of states $\mathbf{s}^t$ and actions $\mathbf{a}^t$) and a new environment where the robot collaborates with the person and help achieve the same goal as quickly as possible. During training, the ground-truth goal of the person is shown to the robot as supervision; during testing, the robot no longer has access to the ground-truth goal and thus has to infer it from the given demonstration.

### 3.2 LOGIC-BASED MODEL

We regard the trajectories and actions of each agent as a sequence of events. In general, the logic rule in our paper is defined as a logical connection of predicates, including property predicates and spatial-temporal relation predicates. The *property* predicates illustrate some intrinsic properties of objects, such as "move slowly", and the spatial-temporal *relation* predicates (*e.g.* "book" is inside the "bookshelf") are introduced to define the spatial and temporal relations of two entities. For example, a sensible rule will look like:

$$
\begin{aligned}
Watch(person, TV) \leftarrow &Walk\_to(person, bedroom) \bigwedge Switch\_on(person, TV) \\
&\bigwedge Sit\_on(person, sofa) \bigwedge Move\_slowly(person)
\end{aligned}
\tag{1}
$$

which means that a person walked slowly into the bedroom, switched on the television, and sat on the sofa. From this, we can infer that the person intends to watch TV. This logical rule may seem simple, but it applies to various different scenarios because most people, when wanting to watch TV, go through these stages.

We aim to develop a model that is informed by logical reasoning, with the capability to predict and rationalize the agent's actions. This model utilizes a query vector $\mathbf{v} = (a, s)$, comprising the action $a$ and the state $s$, along with the historical trajectory $\tau_t$ observed up to the current timestep $t$. The integration of these elements enables our model to provide insightful explanations for the agent's behavior within the given context. Assume that for each possible symbolic state $g \in \mathcal{G}$, where $g$ serves as the *head predicate* and $\mathcal{G}$ represents the set of head predicates, we propose to infer the

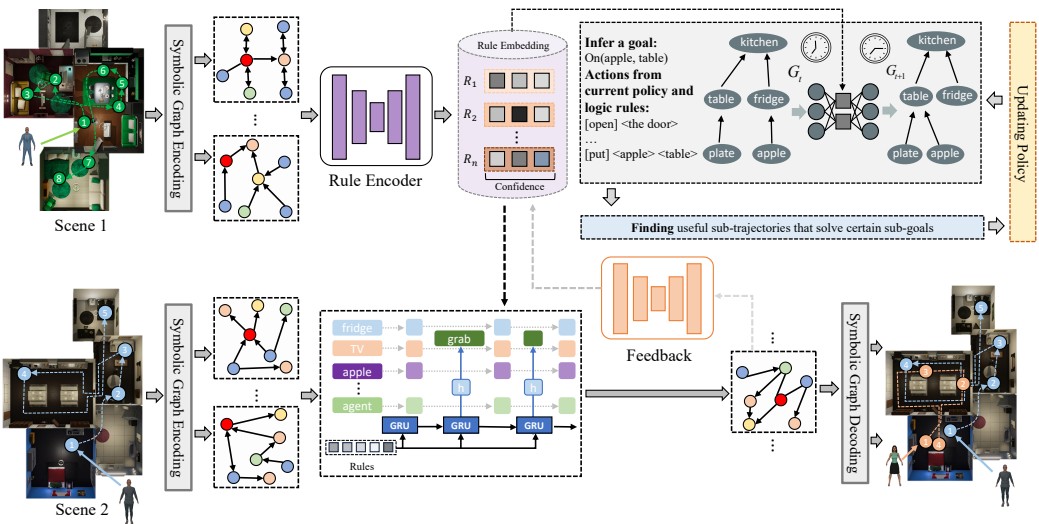

Figure 2: The overview of our proposed framework. In the first stage, given the historical motion trajectory of a person on a scene, we first utilize the symbolic graph to represent objects as nodes and the spatial relationship between them as edges. The rule encoder generates logic rules with their scores to infer the goal iteratively that the person was trying to achieve. In the second stage, the AI agent is asked to work together with the person to achieve the same goal in a new environment. The new scene is also encoded as graphs, which will pass through the propagation module to generate the AI agent's actions based on its latest observation and logic rules.

predicate that is conditional on history and query as $\xi(g \mid \mathbf{v}, \tau_t)$. Suppose there is a rule set $\mathcal{F}_g$, and all rules will play together to reason about the occurrence of $g$. For each $f \in \mathcal{F}_g$, one can compute the features as above. Given the rule set $\mathcal{F}_g$, we model the probability of the event $g$ as follows:

$$p_\alpha(g \mid \mathbf{v}, \tau_t) \propto \exp\left(\sum_{f \in \mathcal{F}_g} \alpha_f \cdot \xi_f(g \mid \mathbf{v}, \tau_t)\right), \tag{2}$$

We optimize the logic-based model by maximizing the likelihood of data as $\mathbb{E}_{(g, \mathbf{v}, \tau_t)}\left[\log \mathbb{E}_{p_\theta}\left[p_\alpha\left(g \mid \mathbf{v}, \tau_t\right)\right]\right]$, where $\theta$ and $\alpha$ are learning weights. The detailed learning process and graph propagation module can be found in the Appendix. Then, we can use these interpretable logic rules to reason the actions of agents in the ToM collaboration framework.

### 3.3 HIERACHYCAL TOM COLLABORATION FRAMEWORK

**Iterative Goal Inference** By leveraging the trajectory information up to the current moment, we can derive the probability distribution for the next potential predicate state, expressed as:

$$g^* = \arg\max_g p_\alpha(g|s^t, a_H^t, \tau^t) = \arg\max_g \frac{\exp\left(\sum_{f \in \mathcal{F}_g} \alpha_f \cdot \xi_f(g \mid s^t, a_H^t, \tau^t)\right)}{\sum_{g'} \exp\left(\sum_{f \in \mathcal{F}_{g'}} \alpha_f \cdot \xi_f(g' \mid s^t, a_H^t, \tau^t)\right)}. \tag{3}$$

Here, $g^*$ denotes the most probable goal inferred by the robot, which both the robot and the human strive to achieve within the next time horizon. Additionally, $a_H^t$ signifies the human action at time $t$. $\mathcal{F}_g$ represents all possible rules in the rule set, and $\sum_{g'} \exp\left(\sum_{f \in \mathcal{F}_{g'}} \alpha_f \cdot \xi_f(\cdot)\right)$ is the partition function used to aggregate the probabilities of paths leading to the same goal.

The primary objective of this collaborative framework between the human and the robot is to identify the policy function $\pi$ for the robot and assist people in accomplishing tasks within their capabilities. Additionally, the robot is required to consider the human's cognitive processes and construct a mental model of the human. This model enables the robot to anticipate the human's policy and enhance collaboration through deeper understanding and consideration.

**Policy Estimation**    To formulate the policies for both the robot agent and the human agent model in the robot's mind, we employ the Boltzmann rationality model, where $\beta$ serves as the rationality coefficient for the agent. This coefficient quantifies the extent to which the agent's choices are focused around the optimal decisions. As $\beta$ approaches infinity, the agent behaves in a manner akin to a perfectly rational agent, meticulously optimizing its choices. Conversely, as $\beta$ approaches 0, the agent becomes increasingly indifferent to the quality of $Q$. The collaboration is purely decentralized, with all agents acting based solely on their local observations and without any direct communication, except for ToM inferences from trajectory information. So the policy function for the agent $i$ can be expressed as follows:

$$\pi_{g,i}(a \mid s^t, \omega^t, \tau^t) = \frac{\exp(\beta \sum_{\omega \sim \Omega} b^t(\omega_{-i} \mid \tau^t) Q_g(a, s^t))}{\sum_{a' \in A} \exp(\beta \sum_{\omega \sim \Omega} b^t(\omega_{-i} \mid \tau^t) Q_g(a', s^t))}. \tag{4}$$

Here, $b^t(\omega_{-i} \mid \tau^t)$ represents the agent's belief, a probability distribution used to estimate the private state $\omega_{-i}$ of the other agent based on this agent's trajectory. $Q_g(a, s^t)$ is the Q-function, dependent on the state and action, and can be represented as:

$$Q_g(a, s^t) = \sum_{f \in \mathcal{F}_g} \alpha_f \cdot \xi_f(g \mid a, s^t, \tau_t). \tag{5}$$

It is worth noting that our policy function and Q function are all based on the current goal $g$ in the rule $f_g \in \mathcal{F}$. In fact, one policy and Q-value for each subtask in the hierarchy: $\boldsymbol{\pi} = \{\pi_{g_0}, \ldots, \pi_{g_m}\}$ and $\boldsymbol{Q} = \{Q_{g_0}, \ldots, Q_{g_m}\}$. In each subtask (one rule in the rule set), the robot and the human need to take a set of primitive actions to reach the terminal states specified by this rule. Then, the next goal can be inferred and is conditioned on the current history and the next primitive action that either the human or the robot is expected to take. Our approach employs a hierarchical policy execution strategy, akin to standard programming languages that use a stack discipline. Each subtask policy receives a state as input and returns the name of a subtask to invoke. When a subtask is invoked, its name is pushed onto the Task-Stack, and its policy is executed until it reaches one of its terminal states. Once a subtask terminates, its name is popped off the Task-Stack. Importantly, if any subtask currently on the Task-Stack terminates, all subtasks below it are immediately aborted. Control then returns to the subtask that initially invoked the terminated subtask. This hierarchical approach enables the dynamic execution of subtasks and completes the final goal in a manner like online inference.

**Belief Update**    However, in a ToM collaboration framework, the Q-function of each agent model should include two additional arguments to account for both agents' intentions and the estimated belief of the counterpart. The intentions of these two agents are considered as prior knowledge and are drawn from $\boldsymbol{\Omega} = \{\Omega_R, \Omega_H\}$ respectively, where $\Omega_R$ represents the robot's intention, and $\Omega_H$ represents the human's intention in the robot's mind. Besides, we employ the obverter technique (Choi et al., 2018) to enable both agent models to hold a belief $\hat{b}_{-i}$ as an estimation of the other agent's belief about their own intention $\omega$. To model this nested belief, we do not need to use a distribution over the original intention distribution. This is because the belief update process follows a deterministic path for rational agents when employing Bayesian inference. At each time step, the update process can be expressed as:

$$b^t(\omega_{-i} \mid \tau^t) = P(\omega_{-i} \mid \tau^{t-1}, a^t) \propto \hat{\pi}(a^t_{-i} \mid s^t, \omega_{-i}) b^{t-1}(\omega_{-i}) \tag{6}$$

In this equation, $\hat{\pi}$ represents the agent's estimation of the other agent's policy, which is learned during centralized training. Our belief update function incorporates counterfactual reasoning. Essentially, the agent explores all possible private states $\omega_{-i}$ and estimates the likelihood of the actions it observed under the assumption that a specific set of private agent states is correct. This approach involves fixing one agent model while training the other, thus avoiding non-stationarity issues that could arise from simultaneous agent updates. Consequently, the other agent model can be considered as part of the environment, ensuring the convergence of the Q-function of the learning agent. The whole algorithm is presented in Algorithm 1, where $\psi_{(i,Q)}, \psi_{(i,\pi)}, \psi_{(i,\mathcal{E})}$ represent the parameters of Q-function, policy function, and belief-updated function $\mathcal{E}$, respectively.

**Training**    In Algorithm 1, Deep Q-learning (Mnih et al., 2013) is employed to train the Q-function $Q_i$. Simultaneously, supervised learning is used to train both the policy function $\hat{\pi}_{-i}$ and the belief update function $\mathcal{E}_{-i}$. The variable $\bar{b}_{-i}$, a result of discretizing $b_{-i}$, is provided as supervisory input

---

**Algorithm 1** Adaptive Theory of Mind Collaboration Hierarchical Learning Algorithm

---

Randomly initialized network parameters $\{\boldsymbol{\psi_{(i,Q)}}, \boldsymbol{\psi_{(i,\pi)}}, \boldsymbol{\psi_{(i,\mathcal{E})}}\}$, $\{\hat{\boldsymbol{\psi}}_{(i,Q)}, \hat{\boldsymbol{\psi}}_{(i,\pi)}, \hat{\boldsymbol{\psi}}_{(1,\mathcal{E})}\}$, learning rate $\{\eta_Q, \eta_\pi, \eta_\mathcal{E}\}$

1: **for** each iteration **do**
2:     **for** $i$ in $\{-1, 1\}$ **do**
3:         **for** episode $= 1, \ldots, m$  (where $m$ is the number of sub-goals) **do**
4:             **repeat**
5:                 Sample sub-goal: $g \leftarrow p\left(g \mid s^t, a^t, \tau^t\right)$
6:                 Sample an action $\hat{a}_i^t \leftarrow \hat{\pi}_i\left(a \mid s^t, \omega^t, \tau^t\right)$
7:                 Sample an action $\hat{a}_{-i}^t \leftarrow \hat{\pi}_{-i}\left(a \mid s^t, \omega^t, \tau^t\right)$
8:                 Update two agent models' beliefs as Eq. (6)
9:             **until** Satisfy the sub-goal
10:         **end for**
11:         $y_i^{t,(k)} = r_i^{t,(k)} + \gamma \max_{a \in A_i} \hat{Q}_{\hat{\psi}_{(i,Q)}}(a, s^{t+1,(k)}, \omega^{(k)}, \hat{b}_i^{t,(k)})$ ($k$ is the index of trajectory)
12:         $L^Q = \sum_t ||Q_{\psi_{(i,Q)}}(a_t^{(k)}, s_t^{(k)}, \omega^{(k)}, \hat{b}_t^{(k)}) - y_t^{(k)}||^2$ Compute Q-loss
13:         $L^\pi = \sum_{t,(k)} H(\hat{\pi}_{(-i,g)}(a_{-i}^t | s^{t,(k)}, \omega_{-i}^{t,(k)}, \tau^{t,(k)}), a_{-i}^{t,(k)})$ Compute policy loss
14:         $L^\mathcal{E} = \mathrm{KL}(\bar{b}_{-i}^{t,(k)} || \mathcal{E}_{(-i,g)}(\hat{b}_{-i}^{t-1,(k)}, a_i^{t,(k)}, s^{t,(k)}))$ Compute belief loss
15:

$$\{\boldsymbol{\psi_{(i,Q)}}, \boldsymbol{\psi_{(i,\pi)}}, \boldsymbol{\psi_{(i,\mathcal{E})}}\} \leftarrow \{\boldsymbol{\psi_{(i,Q)}}, \boldsymbol{\psi_{(i,\pi)}}, \boldsymbol{\psi_{(i,\mathcal{E})}}\} - \nabla_{\{\boldsymbol{\psi_{(i,Q)}}, \boldsymbol{\psi_{(i,\pi)}}, \boldsymbol{\psi_{(i,\mathcal{E})}}\}} \{\eta_Q L^Q, \eta_\pi L^\pi, \eta_\mathcal{E} L^\mathcal{E}\}$$

16:         Update $\{\hat{\boldsymbol{\psi}}_{(i,Q)}, \hat{\boldsymbol{\psi}}_{(i,\pi)}, \hat{\boldsymbol{\psi}}_{(i,\mathcal{E})}\} \leftarrow \{\boldsymbol{\psi_{(i,Q)}}, \boldsymbol{\psi_{(i,\pi)}}, \boldsymbol{\psi_{(i,\mathcal{E})}}\}$ for Q-learning
17:     **end for**
18: **end for**

---

to agent $i$ streamlining the training of the belief update function $\mathcal{E}_{-i}$. Three loss terms are optimized sequentially. $L^Q$ quantifies the disparity between Q-values, while the second term, $L^\pi$, corresponds to the softmax cross-entropy loss. The third term, $L^\mathcal{E}$, measures the Kullback-Leibler (KL) divergence used to assess the dissimilarity between the predicted partner's belief and the supervised partners' belief $\bar{b}_{-i}$.

## 4 EXPERIMENT

In this section, we provide some implementation details and show ablation studies to evaluate the performance of our framework on the Watch-and-help dataset (Puig et al., 2020) and Hand-MeThat dataset (Wan et al., 2022). We compare our model with several state-of-the-art approaches in human-AI interaction, including WAH (Puig et al., 2020), GEM (Netanyahu et al., 2022), DRRN (Hausknecht et al., 2020), Seq2Seq (Sutskever et al., 2014) and ReQueST (Reddy et al., 2020).

For the Watch-and-help (Puig et al., 2020) dataset, we first provide Bob a video that shows Alice successfully performing the activity (Watch stage), and then place both agents in a new environment where Bob has to help Alice achieve the same goal with the minimum number of time steps (Help stage). We utilize success rate, average number of movements in successful episodes, and Speedup as our metrics. Speedup means we compare the episode length when the human and AI agent are working together with the episode length when the human is working alone. For the HandMeThat dataset, we utilize the average score, the success rate(%) that the model achieves the goal within 40 steps, and the average number of moves to evaluate the performance. We split it into four levels depending on the episode length, and run each testing task for five times using different random seeds. For HandMeThat dataset (Wan et al., 2022), It contains 14 locations and typically more than 200 movable objects, which induces a large set of possible actions. We also split it into four hardness levels, and the gaps between levels correspond to different challenges.

Table 1: Comparative performance of various rule encoder backbones on the Watch-and-Help Dataset. An evaluation across four difficulty tiers using three metrics—Average Number of Moves (AN, lower is better) for successful episodes, Success Rate (SR, higher is better), and Speedup (SU, higher is better).

| Methods | | Partially Observable | | | |
|---|---|---|---|---|---|
| | | Level 1 | Level 2 | Level 3 | Level 4 |
| WAH | AN ↓ | 15.28±0.06 | 22.24±0.15 | 28.88±0.15 | 51.20±0.12 |
| | SR ↑ | 78.61±0.02 | 71.02±0.08 | 60.13±0.11 | 51.48±0.04 |
| | SU ↑ | 0.17±0.012 | 0.14±0.019 | 0.13±0.016 | 0.08±0.016 |
| GEM | AN ↓ | 14.08±0.04 | 20.40±0.10 | 28.71±0.05 | 42.12±0.04 |
| | SR ↑ | 79.27±0.02 | 71.32±0.04 | 60.99±0.07 | 53.97±0.06 |
| | SU ↑ | 0.24±0.017 | 0.22±0.016 | 0.19±0.012 | 0.15±0.015 |
| DRRN | AN ↓ | 13.47±0.13 | 20.10±0.14 | 28.68±0.15 | 38.25±0.01 |
| | SR ↑ | 80.86±0.09 | 71.74±0.06 | 66.27±0.13 | 57.53±0.18 |
| | SU ↑ | 0.25±0.015 | 0.19±0.017 | 0.17±0.004 | 0.16±0.009 |
| Seq2Seq | AN ↓ | 12.76±0.06 | 19.92±0.06 | 28.32±0.02 | 34.16±0.04 |
| | SR ↑ | 81.12±0.01 | 72.45±0.18 | 66.69±0.18 | 57.86±0.09 |
| | SU ↑ | 0.38±0.003 | 0.28±0.008 | 0.26±0.007 | 0.21±0.014 |
| ReQueST | AN ↓ | 12.68±0.03 | 18.72±0.10 | 28.12±0.12 | 32.92±0.18 |
| | SR ↑ | 81.12±0.05 | 77.44±0.17 | 68.52±0.04 | 55.34±0.07 |
| | SU ↑ | 0.38±0.019 | 0.30±0.015 | 0.25±0.009 | 0.22±0.011 |
| Ours | AN ↓ | 11.40±0.04 | 16.21±0.15 | 26.16±0.02 | 31.10±0.05 |
| | SR ↑ | 82.43±0.17 | 78.14±0.03 | 69.84±0.17 | 58.91±0.16 |
| | SU ↑ | 0.48±0.015 | 0.44±0.009 | 0.36±0.013 | 0.33±0.017 |

## 4.1 QUANTITATIVE ANALYSIS

We compare our method with several state-of-the-art approaches, and Table 1 presents the qualitative results in the Help stage of the Watch-and-help dataset. Note that WAH (Puig et al., 2020) represents a hybrid model of RL and planning, where regression planning is deployed to generate plans for each sub-goal sampled from the RL-based high-level policy; DRRN (Hausknecht et al., 2020) represents Deep Reinforcement Relevance Network. We observe that our method significantly outperforms all baselines measured by success rate with the fewest number of moves by using the true goals and reasoning about human's future plan, avoiding redundant actions and collisions with him. It exceeds the second-best performance of Seq2Seq by 1.05% and 3.15% in Level 4 and Level 3 respectively. Compared with WAH, we also obtained 7.63% and 9.71% improvements. This is because the rule encoder and evaluator can collaborate with each other to reduce search space and learn better rules. Figure 3 illustrates an example of collaboration. Our framework can actually predict users' intentions and represent some help strategies to achieve the same goal. For the HandMeThat dataset, the performance of learning models in the partially-observable setting is shown in Table 2. Note that the Random model presents an agent that randomly selects a valid action at each time step. Obviously, our model performs decently well on all levels in partially-observable settings.

## 4.2 VISUALIZATION

We show some visual representations of human-robot collaboration on the Watch-and-help dataset in Figure 4. In this picture, Bob's goal is to put the plate into the microwave, and then watch TV in the bedroom. When he walks towards the plate and grabs it, Alice (human-like agent) predicts his goal as "Inside(plate, microwave)", then walks towards the microwave and opens it to help achieve the same goal as quickly as possible, so that Bob can directly put the plate into the microwave. Obviously, this event can be properly represented by the first rule, which use the spatial and temporal relation between Bob and surrounding objects to infer the human goal and help them to finish it.

Table 2: Experiment results on HandMeThat dataset. Each model is evaluated on 4 hardness levels with 3 metrics: the Average Score (AS, higher is better), the Success Rate (SR, higher is better), and the Average Number of Moves (AN, lower is better) in successful episodes.

| Methods | | Partially Observable | | | |
|---|---|---|---|---|---|
| | | Level 1 | Level 2 | Level 3 | Level 4 |
| Random | AS ↑ | -40.0 | -40.0 | -40.0 | -40.0 |
| | SR ↑ | 0.0 | 0.0 | 0.0 | 0.0 |
| | AN ↓ | N/A | N/A | N/A | N/A |
| Seq2Seq | AS ↑ | -5.1±0.07 | -25.3±0.09 | -34.5±0.15 | -32.0±0.20 |
| | SR ↑ | 25.50±0.20 | 10.40±0.14 | 3.95±0.23 | 5.30±0.26 |
| | AN ↓ | 4.21±0.01 | 4.17±0.04 | 4.19±0.02 | 4.12±0.06 |
| GEM | AS ↑ | -6.3±0.11 | -22.5±0.12 | -30.9±0.14 | -29.6±0.18 |
| | SR ↑ | 24.64±0.40 | 12.47±0.21 | 5.74±0.15 | 7.21±0.13 |
| | AN ↓ | 4.34±0.04 | 4.36±0.02 | 4.24±0.00 | 4.22±0.00 |
| DRRN | AS ↑ | -40.0 | -40.0 | -40.0 | -40.0 |
| | SR ↑ | 0.0 | 0.0 | 0.0 | 0.0 |
| | AN ↓ | N/A | N/A | N/A | N/A |
| Ours | AS ↑ | -1.4±0.03 | -5.8±0.06 | -9.1±0.17 | -11.9±0.23 |
| | SR ↑ | 27.73±0.29 | 24.81±0.36 | 20.97±0.12 | 21.66±0.18 |
| | AN ↓ | 4.05±0.01 | 4.09±0.01 | 4.14±0.03 | 4.21±0.02 |

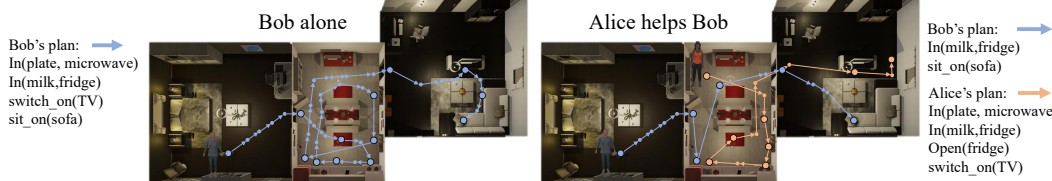

Figure 3: The arrows indicate moving directions and the circles with black borders indicate moments when agents interacted with objects. When working alone (left), Alice had to search different rooms; but with Bob's help (right), Alice could finish the task much faster.

Next, Alice sees that Bob is moving into the bedroom and getting close to the sofa, so Alice infers his goal as "Watch(TV)" based on generated logical rules. Then, Alice walks towards the TV and switches on it, so that Bob can sit down immediately and finish his goal without any redundant actions. As we can see, our rule can actually predict users' intentions and represent some help strategies to achieve the same goal with the minimum number of time steps.

## 5 LIMITATION

The present work has a few limitations. First, for real-world applications, we need to manually set up different spatial-temporal property and relation predicates so that it can infer the graph structure for a large number of objects. Also, the quality of generated rules mainly depends on the setting of pre-defined predicates. We can potentially achieve this by using language instructions to guide the inference. For instance, a human user can provide a language description of the task in addition to the physical demonstration to help achieve a better understanding of the task. Another limitation is that currently we only consider chain rules provided as prior knowledge. In the future, we plan to explore more efficient and effective rule reasoning algorithms and consider more complex rules.

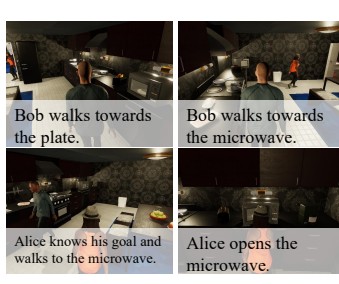

Rule: Walk_to(Bob, plate) ∧ Grab(Bob, plate) ∧ Walk_to(Bob, microwave) ∧ Open(Bob, microwave) →Inside(plate,microwave)
Explanation:
- Bob walks towards the plate and grabs it
- Bob walks towards the microwave
- Alice infers his goal as "put the plate into the microwave"
- Alice opens the microwave

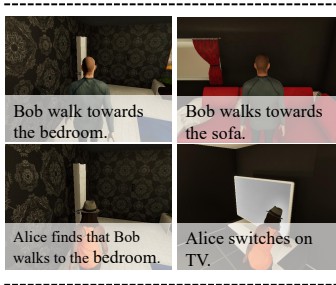

Rule: Walk_to(Bob, bedroom) ∧ Switch_on(Alice, TV) ∧ Walk_to (Bob, sofa) →Watch(Bob, TV)
Explanation:
- Bob walks towards the bedroom
- Bob walks towards the sofa
- Alice infers his goal as "watch TV"
- Alice switches on TV

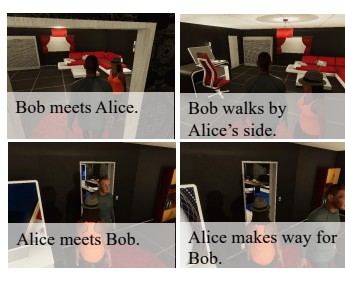

Rule: Walk_to(Bob, sofa) ∧ Block(Alice,Bob) ∧ Make_way (Alice,Bob) →Pass_by(Bob,Alice)
Explanation:
- Bob walks towards the sofa
- Alice blocks Bob's way by chance
- Alice makes way for Bob so he can walk quickly
- Bob walks by Alice's side

Figure 4: Visualization and explanation of logic rules in Watch-and-help dataset.

## 6 CONCLUSION

In this paper, we introduced a novel logic-guided reasoning framework aimed at advancing human-AI collaboration. By infusing logical reasoning into AI agents, our model demonstrates a unique capability to infer and adapt to the goals of others based on observed behaviors. This is achieved by treating logic rules as latent variables and iteratively training the AI agent. In the experiments, our method can analyze the movement sequence of agents in common household activities and obtain novel insights from generated logic rules. Looking ahead, we aim to integrate additional physical laws into our framework, like the principles of energy and momentum conservation. Furthermore, we plan to introduce a predicate generator into our system. This addition will endow AI agents with advanced capabilities, enabling them to generate and reason with new predicates, thereby significantly enriching their understanding and interaction with complex environments.

## 7 ACKNOWLEDGEMENT

Shuang Li's research was in part supported by the NSFC under grant No. 62206236, Shenzhen Science and Technology Program JCYJ20210324120011032, National Key R&D Program of China under grant No. 2022ZD0116004, Shenzhen Key Lab of Cross-Modal Cognitive Computing under grant No. ZDSYS20230626091302006, and Guangdong Key Lab of Mathematical Foundations for Artificial Intelligence.

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
