# Enhancing Human-AI Collaboration Through Logic-Guided Reasoning - Supplementary Material

**Chengzhi Cao[1,2], Yinghao Fu[1], Sheng Xu[1], Ruimao Zhang[1], Shuang Li[1]***
[1]The Chinese University of Hong Kong (Shenzhen)
[2]University of Science and Technology of China
`chengzhicao@mail.ustc.edu.cn, 222051025@link.cuhk.edu.cn,`
`shengxu1@link.cuhk.edu.cn, zhangruimao@cuhk.edu.cn`
`lishuang@cuhk.edu.cn`

## 1 Dataset

**Watch-and-help dataset (Puig et al., 2021).** This dataset consists of a training set with 1011 tasks and 2 testing sets. For each task, we first provide Bob a video that shows Alice successfully performing the activity (Watch stage), and then place both agents in a new environment where Bob has to help Alice achieve the same goal with the minimum number of time steps (Help stage). The helping environment in each task is different from the environment in the pairing demonstration by sampling a different apartment and then randomizing the initial state, while the goals in the test set are unseen during training. During training, we randomly sample one of the 1011 training tasks for setting up a training episode. For evaluating an AI agent on the testing set, we run each testing task for five times using different random seeds and report the average performance. We also split it into four levels, depending on the episode length.

**HandMeThat dataset (Wan et al., 2022).** It contains 14 locations and typically more than 200 movable objects, which induces a large set of possible actions. Each scene contains more than 200 entities with diverse attributes, which resembles a typical real household environment. We also split it into four hardness levels, and the gaps between levels correspond to different challenges.

we split HandMeThat into four hardness levels, and the gaps between levels correspond to different challenges. Recall that $A(G)$ denotes the set of all useful grounded subgoals for goal $G$. $A(m)$ and $A(u)$ are the sets of all possible grounded subgoals for the lifted subgoal $m$ and the utterance $u$. Finally, we consider the subgoal derived from pragmatic reasoning. The four hardness levels are:

**Level 1**: $A(m) = A(u)$. The utterance has no ambiguity. In this case, the instruction understanding task is a pure grounding task: the agent only needs to select the object that satisfies the specification.

**Level 2**: $A(m) = A(u) \bigcap A(G)$. The second level requires social reasoning: the robot can successfully accomplish the task if it can both ground u and infer the human goal G from observations.

**Level 3**: $A(m) = A(r)$. The third level requires all reasoning capabilities combined: the agent need to infer the human goal G. Next, it should make pragmatic reasoning based on G and u to derive r.

**Level 4**: $A(m) \subset A(r)$. In this case, the human utterance u is inherently ambiguous and can not be resolved even with all reasoning capabilities. In this case, further information gathering is needed.

## 2 Difference Between Logic Reasoning and Knowledge Graph

We clarify the differences in the definitions of entity and the relation between logic reasoning and knowledge graph reasoning as follows:

**Definition of Entity.** In logical reasoning, an entity refers to an individual object or thing that exists in a domain of interest. Entities are typically represented by constants or variables in logical

---

*Corresponding author

formulas or expressions. In this context, entities are often abstract and do not necessarily have any explicit semantic meaning. They serve as placeholders or representations for objects or concepts under consideration. In knowledge graph reasoning, an entity represents a specific object, concept, or instance in the real world. In a knowledge graph, entities are typically represented as nodes, and they can have attributes or properties associated with them. These entities are often grounded in real-world entities, such as persons, locations, or organizations. The main difference between entities in logic reasoning and those in knowledge graph reasoning lies in their representation and use within a system. In logical reasoning, entities are abstract and do not necessarily have a distinct correspondence to real-world objects, while entities in knowledge graphs are explicitly linked to real-world objects or concepts and are part of a larger interconnected structure.

**Definition of Relation.** In logical reasoning, a relation represents a connection or association between entities. Relations are often represented by predicates or logical operators that indicate the relationship between the entities. These relations can be unary (relating an entity to itself), binary (relating two entities), or higher-order (relating multiple entities). Relations are often abstract and do not have explicit semantics. In knowledge graph reasoning, Relations are typically represented as edges connecting nodes in the graph. They have specific meanings associated with them, such as "is-a," "part-of," or "located-in." Knowledge graph reasoning involves leveraging these semantic relationships to make inferences, perform queries, or discover new knowledge. The main difference is that relations in logical reasoning can have any number of entities, while relations in knowledge graph reasoning can only tolerate two entities because the unit of knowledge graph is the "Entity-Relation-Entity" triplets.

## 3 OPTIMIZATION

Our rule encoder is deployed as Transformer-based framework, and the distribution of rules can be represented as $\Psi(g|N, \text{Trans}(\mathbf{v}, \tau_t))$, where $\Psi(\cdot)$ is multinomial distributions, $N$ is the number of selected rules, and $\text{Trans}(\mathbf{v}, \tau_t)$ defines a distribution over compositional rules with spatial-temporal states. The generative process of the rule set is quite intuitive, where we simply generate $N$ rules to form $z$. The input of our framework is the historical trajectory of entities, and the output is the generated logic rules. The historical trajectory would be first encoded into a scene graph and pass through three-layer MLPs before being fed into the transformer, and a ReLU non-linearity following each of the first two layers. In the rule encoder, the dimensions of keys, values, and queries are all set to 256, and the hidden dimension of feed-forward layers is 512. The number of heads for multi-head attention is 8.

For feedback module, suppose there is a rule set $\mathcal{F}_g$, where the event $g$ is the head predicate (subgoal). All the rules will play together to reason about the occurrence of $g$. Given the rule set $\mathcal{F}_g$, we model the probability of the event $g$ as a log-linear function:

$$p(g \mid \mathbf{v}, \tau_t) \propto \exp\left(\sum\nolimits_{f \in \mathcal{F}_g} \alpha_f \cdot \phi_f(g \mid \mathbf{v}, \tau_t)\right). \tag{1}$$

Assume that the rule content is represented as latent embeddings $z$, and the feedback can be calculated as:

$$p_\alpha(g|\mathbf{v}, z, \tau_t) = \frac{\exp\left(\sum_{f \in z_g} \alpha_f \cdot \phi_f(g|\mathbf{v}, \tau_t)\right)}{\sum_{g'} \exp\left(\sum_{f \in z_{g'}} \alpha_f \cdot \phi_f(g'|\mathbf{v}, \tau_t)\right)}. \tag{2}$$

In the second stage, the AI agent is asked to work together with the person to achieve the same goal in a new environment as fast as possible. The new scene is also encoded as graphs, which will pass through GRU to generate AI agent's actions at every step based on its latest observation and logic rules. Finally, the feedback module is updated to be consistent with the high-quality rules identified. This module is also transformer-based architecture, where the dimensions of keys, values, and queries are all set to 256, the hidden dimension of feed-forward layers is 512, and the number of heads for multi-head attention is 8.

Each training iteration starts with an update to the rule encoder $p_\theta$, after which we change the feedback module $p_\alpha$ based on certain rules produced by the encoder. Particularly, a Transformer-based

Table 1: Predicate sets used for defining the goal of agents in five types of activities.

| | |
|---|---|
| Put groceries | IN(cupcake, fridge), IN(pancake, fridge), IN(poundcake, fridge), IN(pudding, fridge), IN(apple, fridge), IN(juice, fridge), IN(wine, fridge) |
| Prepare a meal | ON(coffeepot, dinnertable), ON(cupcake, dinnertable), ON(pancake, dinnertable), ON(poundcake, dinnertable), ON(pudding, dinnertable), ON(apple, dinnertable), ON(juice, dinnertable), ON(wine, dinnertable) |
| Set up a dinner table | ON(plate, dinnertable), ON(fork, dinnertable), ON(waterglass, dinnertable), ON(wineglass, dinnertable) |
| Read a book | HOLD(Alice, book), SIT(Alice, sofa), ON(cupcake, coffeetable), ON(pudding, coffeetable), ON(apple, coffeetable), ON(juice, coffeetable), ON(wine, coffeetable) |
| Wash dishes | IN(plate, dishwasher), IN(fork, dishwasher), IN(waterglass, dishwasher), IN(wineglass, dishwasher) |

encoder transform the observed action trajectories to the latent rule space, yielding the posterior probabilities of the latent rule $z$. Following Cao et al. (2023), each candidate rule is created sequentially in the latent rule space, and the posterior probability of each rule sequence can be assessed.

In order to predict $g$, we draw many rules $\hat{z}$ for each query when optimizing the feedback module. Our goal is to select some high-quality rules $z_I$ ($z_I \subset \hat{z}$, $|z_I| = K$) from all created rules for each query. This is achieved by considering the posterior probabilities of each subset of logic rules $z_I$, with likelihood from the feedback module $p_\alpha$ and prior from the rule encoder $p_\theta$. $\log p_{\theta,\alpha}(z_I|\mathbf{v}, \tau_t)$ is an approximation for the log-probability, which can be approximated as $\log p_{\theta,\alpha}(z_I|\mathbf{v}, \tau_t)$.

The quality of candidate rules is calculated by subtracting a rule's contribution to the correct event type from its average contribution to the other candidate responses. If a rule receives a greater score for the right kind of occurrence and a lower score for other possible predictions, it is more meaningful. After obtaining many high-quality rules from training data, we use these rules to update the rule encoder by maximizing the log-likelihood as $\sum_{z^{(i)} \in z_I} \log \text{Trans}_\theta(\mathbf{v}, \tau_t) + \text{const}$. The rule encoder will narrow the search space and yield more accurate empirical results as it gains proficiency in producing high-quality rules.

## 4  GRAPH PROPAGATION

Following Liao et al. (2019), we regard the collaboration task as a seq2seq problem, where an encoder encodes the input sketch and the decoder generates the AI agent's action at a time. To do that, we encode the scene as a graph $G = (\mathcal{V}, \mathcal{R})$ modeling the dependencies of the object instances. The node $v \in \mathcal{V}$ indicates the object instance and each node has a label, including the object class $c_v$, its states $l_v$, and properties $prop_v$. Note that $\mathcal{V}$ includes a node for the agent itself. The edge $r \in \mathcal{R}$ encodes the spatial relations.

We adopt gated graph sequence neural network (Li et al., 2015) to obtain the hidden states of the nodes and capture the object relations in the scene graph. The hidden states of each node $v$ are initialized by its label $(c_v, l_v, prop_v)$:

$$h_v^0 = tanh(g_{init}([W_c c_v, W_l l_v, W_{prop} prop_v])), \tag{3}$$

where $W_c, W_l, W_{prop}$ are learnable weights, and $g_{init}$ is a network composed of fully connected layers that combine all the information. At propagation step $k$, each node's incoming information $x_v^k$ is determined by aggregating the hidden states of its neighbors $v' \in \mathcal{N}(v)$ at the previous step $k-1$:

Table 2: Experiment results of different backbones (rule encoder) on Watch-and-help dataset. Each model is evaluated on 4 hardness levels with 3 metrics: the Average Number of Moves (AN, lower is better) in successful episodes, the Success Rate (SR, higher is better), and Speedup (SU, relative reduction in episode length when collaboration or not, higher is better).

| Methods | | Partially Observable | | | |
| --- | --- | --- | --- | --- | --- |
| | | Level 1 | Level 2 | Level 3 | Level 4 |
| CNN | AN $\downarrow$ | 11.59±0.17 | 16.74±0.09 | 27.05±0.12 | 31.95±0.16 |
| | SR $\uparrow$ | 74.40±0.10 | 73.16±0.12 | 69.31±0.18 | 58.03±0.06 |
| | SU $\uparrow$ | 0.46±0.003 | 0.39±0.020 | 0.34±0.019 | 0.30±0.018 |
| GNN | AN $\downarrow$ | 12.47±0.16 | 16.52±0.01 | 26.38±0.09 | 32.07±0.15 |
| | SR $\uparrow$ | 81.01±0.01 | 69.99±0.18 | 69.30±0.11 | 58.77±0.03 |
| | SU $\uparrow$ | 0.44±0.002 | 0.37±0.007 | 0.33±0.007 | 0.27±0.008 |
| LSTM | AN $\downarrow$ | 11.78±0.01 | 16.71±0.09 | 26.46±0.07 | 31.58±0.06 |
| | SR $\uparrow$ | 79.64±0.04 | 74.59±0.02 | 69.54±0.10 | 58.31±0.01 |
| | SU $\uparrow$ | 0.46±0.013 | 0.36±0.004 | 0.31±0.015 | 0.23±0.006 |
| Ours | AN $\downarrow$ | **11.40±0.04** | **16.21±0.15** | **26.16±0.02** | **31.10±0.05** |
| | SR $\uparrow$ | **82.43±0.17** | **78.14±0.03** | **69.84±0.17** | **58.91±0.16** |
| | SU $\uparrow$ | **0.48±0.015** | **0.44±0.009** | **0.36±0.013** | **0.33±0.017** |

$$x_v^k = \sum_{j \in L(\mathcal{R})} \sum_{v' \in \mathcal{N}_j(v)} W_{p_j} h_{v'}^{k-1} + b_{p_j}, \tag{4}$$

where $L(\mathcal{R})$ denotes the set of edge labels and the linear layer $W_{p_j}$ and bias $b_{p_j}$ are shared across all nodes. After aggregating the information, the hidden states of the nodes are updated through a gating mechanism similar to Gated Recurrent Unit (GRU) as follows:

$$\begin{aligned}
z_v^k &= \phi(W_z x_v^k + U_z h_v^{k-1} + b_z), \\
r_v^k &= \phi(W_r x_v^k + U_r h_v^{k-1} + b_r), \\
\hat{h}_v^k &= tanh(W_h x_v^k + U_h(r_v^k \cdot h_v^{k-1}) + b_h), \\
h_v^k &= (1 - z_v^k) \cdot h_v^{k-1} + z_v^k \cdot \hat{h}_v^k,
\end{aligned} \tag{5}$$

which results in a vector embedding for each object $h_v^k$, with information about its state and relationship with the environment.

## 5 BACKBONE

We added the relevant ablation experiments on the different components of the approach. For the rule encoder and the feedback module, we compare ours (transformer-based) with three widely used backbones, including CNN, RNN and GNN (graph neural network), and evaluate them in the Watch-and-help dataset. As shown in Table 2, our architecture can actually achieve superior results in all metrics.

## 6 PREDICATE SETS FOR GOAL DEFINITIONS

Table 3 summarizes the five predicate sets used for defining goals. Note that VirtualHome supports more predicates for potential future extensions on the goal definitions.

## 7 ADDITIONAL RESULTS ON OVERCOOKD GAME

We consider incorporating additional robust human-AI collaboration tasks, including overcooked game, which is a benchmark environment for fully cooperative human-AI task performance. The

Table 3: Experiment results of different backbones (feedback module) on Watch-and-help dataset. Each model is evaluated on 4 hardness levels with 3 metrics: the Average Number of Moves (AN, lower is better) in successful episodes, the Success Rate (SR, higher is better), and Speedup (SU, relative reduction in episode length when collaboration or not, higher is better).

| Methods | | Partially Observable | | | |
|---------|---|---------|---------|---------|---------|
| | | Level 1 | Level 2 | Level 3 | Level 4 |
| CNN | AN↓ | 11.41±0.20 | 16.60±0.08 | 26.79±0.04 | 31.78±0.09 |
| | SR↑ | 72.79±0.17 | 72.27±0.16 | 69.33±0.13 | 58.16±0.06 |
| | SU↑ | 0.44±0.005 | 0.34±0.002 | 0.35±0.030 | 0.27±0.007 |
| GNN | AN↓ | 11.73±0.02 | 16.74±0.18 | 26.23±0.14 | 31.60±0.13 |
| | SR↑ | 81.33±0.06 | 76.29±0.08 | 61.36±0.02 | 57.98±0.14 |
| | SU↑ | 0.37±0.010 | 0.33±0.007 | 0.33±0.013 | 0.25±0.016 |
| LSTM | AN↓ | 12.34±0.11 | 17.08±0.16 | 26.41±0.17 | 31.92±0.15 |
| | SR↑ | 73.24±0.08 | 73.01±0.17 | 62.38±0.11 | 58.30±0.02 |
| | SU↑ | 0.42±0.003 | 0.31±0.017 | 0.32±0.020 | 0.29±0.007 |
| Ours | AN↓ | **11.40±0.04** | **16.21±0.15** | **26.16±0.02** | **31.10±0.05** |
| | SR↑ | **82.43±0.17** | **78.14±0.03** | **69.84±0.17** | **58.91±0.16** |
| | SU↑ | **0.48±0.015** | **0.44±0.009** | **0.36±0.013** | **0.33±0.017** |

Table 4: Rewards over trajectories of 400 timesteps for our methods and Seq2Seq.

| Layouts | Seq2Seq | Ours |
|---------|---------|------|
| Cramped Room | 133.2+8.1 | 158.3+7.1 |
| Asymmetric Advantage | 169.2+3.0 | 185.7+5.4 |
| Coordination Ring | 115.4+7.9 | 142.1+7.1 |
| Forced Coordination | 71.2+4.9 | 85.6+5.7 |
| ACounter Circuit | 60.9+4.4 | 89.0+3.5 |

goal of the game is to deliver soups as fast as possible. Each soup requires placing up to 3 ingredients in a pot, waiting for the soup to cook, and then having an agent pick up the soup and delivering it. The agents should split up tasks on the fly and coordinate effectively in order to achieve high reward. The six possible actions are: up, down, left, right, noop, and "interact", which does something based on the tile the player is facing, e.g. placing an onion on a counter. Each layout has one or more onion dispensers and dish dispensers, which provide an unlimited supply of onions and dishes respectively. This game includes five layouts:

(*1*) **Cramped Room** presents low-level coordination challenges: in this shared, confined space it is very easy for the agents to collide.

(*2*) **Asymmetric Advantages** tests whether players can choose high-level strategies that play to their strengths.

(*3*) **Coordination Ring** players must coordinate to travel between the bottom left and top right corners of the layout.

(*4*) **Forced Coordination** instead removes collision coordination problems, and forces players to develop a high-level joint strategy, since neither player can serve a dish by themselves.

(*5*) **Counter Circuit** involves a non-obvious coordination strategy, where onions are passed over the counter to the pot, rather than being carried around.

As good coordination between teammates is essential to achieve high returns in this environment, we use cumulative rewards over a horizon of 400 timesteps for our agents as a proxy for coordination ability. For all DRL experiments, we report average rewards across 100 rollouts and standard errors across 5 different seeds. We present quantitative results in these five layouts in the following table. There is a large gap between our method and Seq2Seq in all layouts.