# OpenReview forum: "Enhancing Human-AI Collaboration Through Logic-Guided Reasoning"
_ICLR.cc/2024/Conference — ICLR 2024 poster_

### Official Review · Reviewer_dixh · 2023-10-28

**Soundness:** 3 good
**Presentation:** 3 good
**Contribution:** 3 good
**Rating:** 8
**Confidence:** 3

**Summary:**

The paper proposes a framework to engage human-AI collaboration by incorporating logical reasoning aiming to develop a machine to help humans work together. In the framework, trajectories of human activities are given, and logic rules that map spatial-temporal states to goals. Then, the agents are deployed to new environments where they need to work together with humans and are trained to help humans predict their goals from the sequence of actions and states. In the experiments, the proposed framework outperformed other SOTA approaches, e.g., in the success rate. Different datasets/tasks and evaluation metrics are used in the evaluation.

**Strengths:**

The paper addresses an important problem of integrating robots to work with humans. The idea of integrating logical reasoning for generalization in this problem setting seems novel.
The paper provides good empirical evaluations showing the proposed framework’s advantages against baselines.
The experiments are conducted on practical datasets and environments, they give a great insight into the applications of neuro-symbolic methods, where symbolic logic and neural networks are integrated. Limitations are properly discussed.

**Weaknesses:**

Although I appreciate the paper’s ideas and evaluations, I found some concerns about the paper.

First, more discussions comparing the proposed approach to related studies need to be provided. Otherwise, it is not easy to understand how we can distinguish the proposed method from existing frameworks, and providing them would help readers understand the literature clearly.

Moreover, some parts of the method explanation are somewhat hard to follow. In Sec 2.3, the query is introduced as $\mathbf{v} = (\mathbf{a}$, $\mathbf{s})$, but there is no specification for these variables. I suppose this is a tuple of a sequence of actions and a sequence of states, but that should be noted explicitly. An intuitive explanation of what $\mathbf{v}$ stands for would help readers.

I list some minor comments:
- In Eq. (5). it is written as $g \in f$, but I’m not sure this is a standard notation because here $f$ is a rule, not a set.
- Typo: double periods in the caption of Fig. 3.
-  I would kindly suggest including the literature on neuro-symbolic research (e.g. [1,2], but not necessary; it is the author’s choice), since the proposed approach is significantly related to the field, and the community would benefit from this work. The readability of the method explanation could be improved.

[1] Artur d'Avila Garcez, Luís C. Lamb: Neurosymbolic AI: the 3rd wave. Artif. Intell. Rev. 56(11): 12387-12406 (2023)

[2] Henry Kautz. The Third AI Summer: AAAI Robert S. Engelmore Memorial Lecture. AI Magazine, 43(1), 105-125 (2022).

**Questions:**

What are the most related studies? How is the proposed approach compared and superior to them?

Including the time and location in the predicate (in Eq. (4)) would generate a large number of ground atoms (atoms without variables), and typically, reasoners (including probabilistic ones e.g. ProbLog and Markov Logic Networks) need to compute ground atoms to perform reasoning.
Does the proposed framework suffer from the large number of logic representations to be generated in the inference? If not, how does it avoid this problem? Are there any restrictions over the considered language/environments?

---

> ### Author Response · Authors · 2023-11-21
> **Rebuttal by Authors**
>
> **Q1:More discussions comparing the proposed approach to related studies need to be provided.**
>
> A1:In our original submission, **because of the page limitation in the main paper, the description of the related work is provided in the supplementary material Section 2**. It includes a comprehensive overview of the literature (including behavior understanding, logical reasoning, and human-robot interaction) related to our work. We apologize for not including the related work directly in the main document and have added it into our revised manuscript.
>
> **Q2: Moreover, some parts of the method explanation are somewhat hard to follow.**
>
> A2: Thank you for your feedback. The query, denoted as v=(a, s), actually represents a tuple consisting of an action and a state. We have added specifications for these variables in a revision.
>
> **Q3:I list some minor comments.**
>
> A3: Thank you for your suggestion. In our formulation, although f represents a rule, it is also conceptualized as a set comprising all the predicates that constitute the rule. This dual characterization allows us to employ set notation to encapsulate the composite elements of the rule f. Moreover, we also add [1,2] in the related work and analyze their novelty clearly.
>
> [1] Artur d'Avila Garcez, Luís C. Lamb: Neurosymbolic AI: the 3rd wave. Artif. Intell. Rev. 56(11): 12387-12406 (2023)
>
> [2] Henry Kautz. The Third AI Summer: AAAI Robert S. Engelmore Memorial Lecture. AI Magazine, 43(1), 105-125 (2022).
>
> **Q4: What are the most related studies?**
>
> A4: In Section 3.2, we describe several related studies (including WAH, which is the most related study) and provide a brief summary of the novelty of these methods. Moreover, in our experiment, we also compare our methods with WAH, and the results are shown in Table 1 in the main paper. We observe that our method significantly outperforms all baselines measured by success rate with the fewest number of moves.
>
> **Q5:Does the proposed framework suffer from the large number of logic representations to be generated in the inference? If not, how does it avoid this problem? Are there any restrictions over the considered language/environments?**
>
> A5: Yes, the proposed framework suffers from the large number of logic representations to be generated. The best set of logic rules is approximately obtained by sampling and preserving the top-K rules according to their posterior probabilities. **Note that in the supplementary material section 5**, we have demonstrated the optimization of our proposed algorithm, which clearly describes the calculation of the posterior probabilities of the latent rule and the selection of high-quality rules from a large number of logic representations.  This sampling strategy, which focuses on the most probable logic rules, allows our framework to operate efficiently without compromising the integrity of the inference process.

---

> > ### Comment · Reviewer_dixh · 2023-11-22
> >
> > I thank the authors for their response. Since my major concerns have been addressed, I increased my confidence to 3, and keep my score.

---

### Official Review · Reviewer_WpfX · 2023-11-01

**Soundness:** 3 good
**Presentation:** 2 fair
**Contribution:** 2 fair
**Rating:** 5
**Confidence:** 5

**Summary:**

This paper introduces a framework aimed at improving human-robot perception and collaboration by integrating logical rules and Theory of Mind (ToM). Logical rules provide interpretable predictions and generalize well across diverse tasks, making them valuable for learning and decision-making. Leveraging ToM to understand the mental states of others enhances effective collaboration.

In this approach, the authors use logical rules derived from observational data to infer human goals and guide human-like agents. These rules are treated as latent variables, and a rule generator is trained alongside a multi-agent system within the robot's cognitive framework. The process involves two stages: first, assessing the posterior distribution of latent rules using learned embeddings to represent entities and relations, with confidence scores indicating consistency with observed data. Second, a joint optimization of the rule generator and model parameters is performed, maximizing the expected log-likelihood.

**Strengths:**

To assist humans, a hierarchical reinforcement learning model with ToM is employed to plan robot actions.
Multiple experiments validate each component of the framework, and the results on multiple benchmarks demonstrate that this model outperforms the majority of existing approaches.
The combination of logical rules, Theory of Mind, and hierarchical reinforcement learning creates a comprehensive framework for enhancing human-robot collaboration and perception.

**Weaknesses:**

The rule generator and the reasoning evaluator are important but don't show the particular design.
The Iterative Goal Inference seems that a process of conditional learning, so how to implement this part is not clear enough.

**Questions:**

Refer to the above comments.

---

> ### Author Response · Authors · 2023-11-19
> **Rebuttal by Authors**
>
> **Q1:The rule generator and the reasoning evaluator are important but don't show the particular design.**
>
> A1: In our original submission, **because of the page limitation in the main paper**, the design and hyper-parameters of the rule generator and the reasoning evaluator are described **in the supplementary material Section 3, 4, 5 and 6**. Below we show some description for the reviewer’s convenience (more details can be found in the supplementary material).
>
> **Rule Generator:** We deploy Transformer-based framework to model the rule generator $p_{\theta}$. Specifically, we define the distribution of a set of rules as follows:
>
> \begin{align}
> p_\theta(z \mid \mathbf{v}, \tau_t) = \Psi(z|N,\text{Trans}_{\theta}(\mathbf{v}, \tau_t)),
> \end{align}
>
> where $\Psi(\cdot)$ is multinomial distributions, $N$ is the number of the top rules, and $\text{Trans}_\theta(\mathbf{v}, \tau_t)$ defines a distribution over compositional rules with spatial-temporal states. The generative process of the rule set is quite intuitive, where we simply generate $N$ rules to form $z$.
>
> **Reasoning Evaluator:** Suppose there is a rule set $F_{g}$, where the event $g$ is the head predicate (sub-goal). All the rules will play together to reason about the occurrence of $g$. For each $f \in F_{g}$, one can compute the features as above. Given the rule set $F_{g}$, we model the probability of the event $g$ as a log-linear function of the features. Here we assume the rule content is latent, and the rule evaluator is given as:
>
> \begin{align}
> p_\alpha(g | \mathbf{v}, z, \tau_t) = \frac{\exp \left(  \sum\nolimits_{f \in z_{g}} \alpha_{f} \cdot   \phi_{f}(g| \mathbf{v}, \tau_t) \right)}{\sum_{g'}\exp \left(  \sum\nolimits_{f \in z_{{g}'}} \alpha_{f} \cdot   \phi_{f}({g}'| \mathbf{v}, \tau_t) \right)}.
>  \end{align}
>
> **Hyper-parameters:** In the first stage, the input of our framework is the historical trajectory of entities, and the output is the generated logic rules. The historical trajectory would be first encoded into a scene graph and pass through three-layer MLPs before being fed into the transformer, and a ReLU non-linearity following each of the first two layers. In rule generator, the dimensions of keys, values, and queries are all set to 256, and the hidden dimension of feed-forward layers is 512. The number of heads for multi-head attention is 8.
>
> In the second stage, the reasoning evaluator is also transformer-based architecture, where the dimensions of keys, values, and queries are all set to 256, the hidden dimension of feed-forward layers is 512, and the number of heads for multi-head attention is 8.
>
> **Q2:The Iterative Goal Inference seems that a process of conditional learning, so how to implement this part is not clear enough.**
>
> A2: Thanks for your suggestions. Iterative goal inference has been described in Section 2.4, and we make it detailed as follows:
>
> Iterative goal inference is a method that employs the accumulated trajectory data to determine the robot’s most likely goal at any given moment. This process is mathematically represented by the equation $g^* = \arg\max_g p_\alpha(g | s^t, a_H^t, \tau^t)$, where $g^*$ is the inferred goal that maximizes the conditional probability based on the current state $s^t$, the action the human takes $a_{H}^t$ and the trajectory $\tau^t$. To implement Iterative Goal Inference in a practical setting, the robot continually updates its subgoals and beliefs. This is done by integrating the most recent observations and applying predefined logic rules. Such a mechanism permits the robot to craft and execute new plans in real time, thereby enhancing its collaborative capability with humans.
>
> Through a continuous loop of inference based on current data and subsequent action, the robot can effectively align its behavior with human intentions and adapt to changing circumstances in a collaborative environment. Here's a more detailed breakdown of the process:
>
> **Observation Integration**: At each step, the robot integrates new observational data, which includes the current state, human actions, and trajectory information.
>
> **Goal Inference**: Using the probabilistic function, the robot infers the most probable goal at the current moment based on the integrated data.
>
> **Plan Formulation and Execution**: With the inferred goal, the robot formulates a new plan that includes subgoal and action to be taken. It then executes this plan concurrently with the human, allowing for real-time collaborative adjustments.
>
> This iterative process ensures that the robot's assistance is relevant, timely, and contextually appropriate, thereby providing effective support to the human collaborator.

---

> ### Author Response · Authors · 2023-11-22
> **Invitation for further discussion**
>
> Dear Reviewer,
>
> I hope this message finds you well. We've made revisions based on your previous feedback and are keen to know **if there are any remaining issues or further clarifications needed**.
>
> Your insights are very important to us, and we eagerly await your response.
>
> Thank you for your time and effort.
>
> Best regards,
>
> The authors

---

> ### Author Response · Authors · 2023-11-23
> **Response to Reviewer Feedback and Request for Further Guidance**
>
> Thank you for all your constructive comments! **We wonder whether our response has addressed your major concerns to allow the paper to cross your acceptance threshold**. Should there be any further specific issues you would like us to address, we welcome your input and are fully prepared to incorporate any additional modifications into the final version of our manuscript.

---

### Official Review · Reviewer_iG1y · 2023-11-01

**Soundness:** 3 good
**Presentation:** 4 excellent
**Contribution:** 3 good
**Rating:** 6
**Confidence:** 3

**Summary:**

This paper focuses on exploiting logic rules to guide human-like agents. They design a rule generator and rule evaluator to obtain useful rules given entities and relations, and apply hierarchical reinforcement learning with ToM to plan actions. The results show that the model achieves SOTA performance.

**Strengths:**

- The method to construct and utilize knowledge graphs to plan action is novel and interesting.
- The experiments show that the proposed method significantly outperforms baselines in various metrics. With the provided standard deviation, the table is more convincible.

I am not familiar with reinforcement learning and am unable to assess this part.

**Weaknesses:**

- The definitions of entity, relation, and logic rule are inconsistent with the widely agreed definition in the knowledge graph academia. It seems that the users redefine them in the context of their task, while re-using these terminologies of knowledge graph. This makes the paper a little confusing, especially for audiences with a knowledge graph background.
- Different examples of rules are inconsistent. In (3), an item of a logic rule is `Walk_to(person, bedroom)`, but the examples of rule 1 in Fig.3 contain an item `Walk_to(plate)`. It is inconsistent in whether `Walk_to` needs a person as the first entity argument. I wonder which setting is actually used in the method.
- (Minor) In tables, the best results are not easy for audiences to discover. Please use bold text for the best results, and show ↑ or ↓ for each metric to indicate whether greater or less is better.
- (Minor) There are a few confusing wordings. What is the meaning of "hardness level" in the caption of Tab.1 and Tab.2? Its meaning is more like "non-softness" than "difficulty".

**Questions:**

In "Weaknesses".

---

> ### Author Response · Authors · 2023-11-21
> **Rebuttal by Authors**
>
> **Q1:The definitions of entity, relation, and logic rule are inconsistent with the widely agreed definition in the knowledge graph academia.**
>
> A1: Thank you for your valuable suggestions. We clarify the differences in the definitions of entity and the relation between logic reasoning and knowledge graph reasoning as follows:
>
> 1. Definition of Entity
> - In logical reasoning, an entity refers to an individual object or thing that exists in a domain of interest. Entities are typically represented by constants or variables in logical formulas or expressions. In this context, entities are often abstract and do not necessarily have any explicit semantic meaning. They serve as placeholders or representations for objects or concepts under consideration.
>
> - In knowledge graph reasoning, an entity represents a specific object, concept, or instance in the real world. In a knowledge graph, entities are typically represented as nodes, and they can have attributes or properties associated with them. These entities are often grounded in real-world entities, such as persons, locations, or organizations.
> - The main difference between entities in logic reasoning and those in knowledge graph reasoning lies in their representation and use within a system. In logical reasoning, entities are abstract and do not necessarily have a distinct correspondence to real-world objects, while entities in knowledge graphs are explicitly linked to real-world objects or concepts and are part of a larger interconnected structure.
>
> 2. Definition of Relation
>
> - In logical reasoning, a relation represents a connection or association between entities. Relations are often represented by predicates or logical operators that indicate the relationship between the entities. These relations can be unary (relating an entity to itself), binary (relating two entities), or higher-order (relating multiple entities). Relations are often abstract and do not have explicit semantics.
>
> - In knowledge graph reasoning, Relations are typically represented as edges connecting nodes in the graph. They have specific meanings associated with them, such as “is-a,” “part-of,” or “located-in.” Knowledge graph reasoning involves leveraging these semantic relationships to make inferences, perform queries, or discover new knowledge.
>
> - The main difference is that relations in logical reasoning can have any number of entities, while relations in knowledge graph reasoning can only tolerate two entities because the unit of knowledge graph is the “Entity-Relation-Entity” triplets.
>
> We have carefully reviewed the symbol definitions in the paper and distinguished them from the definitions in the knowledge graphs.
>
> **Q2:Different examples of rules are inconsistent.**
>
> A2:Thank you for your valuable feedback. In our setting, the command “Walk_to” needs a person as the first entity argument. We have rectified all ambiguous instructions in the paper.
>
> **Q3:minor errors.**
>
> A3: Following the reviewer’s suggestions, we have modified all tables and added arrows (↑ or ↓) next to each metric to indicate whether a higher or lower value is considered better. These modifications will help readers quickly identify the best results and understand the direction of improvement for each metric.
>
> **Q4:There are a few confusing wordings. What is the meaning of "hardness level" in the caption of Tab.1 and Tab.2? Its meaning is more like "non-softness" than "difficulty".**
>
> A4: We follow [1] to split our datasets into four hardness levels, and the gaps between levels correspond to different challenges. We apologize for any misunderstanding caused by the wording and make the necessary adjustments in the captions of Table 1 and Table 2 to ensure clarity and avoid any further confusion.
>
> [1] Wan Y, Mao J, Tenenbaum J. HandMeThat: Human-Robot Communication in Physical and Social Environments[J]. Advances in Neural Information Processing Systems, 2022, 35: 12014-12026.

---

### Official Review · Reviewer_wN2Q · 2023-11-01

**Soundness:** 2 fair
**Presentation:** 2 fair
**Contribution:** 3 good
**Rating:** 6
**Confidence:** 2

**Summary:**

The paper presents a novel framework aimed at improving human-AI collaboration by integrating logic-guided reasoning and theory of mind. The key contributions include a method for generating and evaluating logic rules from observational data to infer human intentions and actions, a hierarchical reinforcement learning model incorporating theory of mind for planning robot actions to assist humans, and comprehensive experiments on two datasets demonstrating the effectiveness and generalizability of the proposed model.

**Strengths:**

1. This paper proposes a novel method that utilizes logical reasoning to generate a broad understanding of the agent’s objectives in a new environment and thus strengthen social perception and collaboration between humans and AI.
2. The detailed experiment analysis shows the effectiveness of the proposed method.

**Weaknesses:**

1. The experimental tasks seem somewhat weak, as even Seq2Seq (Sutskever et al., 2014) can achieve reasonably good results (Table 1). Could the authors provide some more robust human-AI collaboration tasks for reference? (I apologize for not being familiar with the relevant field, so I can only provide a general reference [1]). Additionally, as a human-AI collaboration method, it would be more convincing if a human study could be included if possible.
2. The authors should provide the related work in the main text rather than in the appendix. The related work can help readers who are not familiar with this field quickly gain background information about this work and its positioning. Furthermore, I recommend adding a section in the related work that focuses on human-AI collaboration, as it is highly relevant to the topic of this paper.
3. Presentation: (a) I suggest that the authors highlight the best results in Table 1 and clarify in the caption whether each metric is better when higher (success rate) or lower (average number of moves). (b) Watch-and-help is cited twice in references. (c) Citation format like WAH Puig et al. (2020) should be WAH (Puig et al., 2020). Use `\citep` command in latex.

[1] Building Cooperative Embodied Agents Modularly with Large Language Models

**Questions:**

Although the author has demonstrated in the experiments that the method in the paper is indeed effective, one point still puzzles me: I understand how theory of mind can assist in improving the effectiveness of human-AI collaboration, but why does logical reasoning help enhance human-AI collaboration? Is it merely because logical reasoning is effective in all similar tasks (i.e. reasoning) rather than just theory of mind or human-AI collaboration? If it is effective in both, why not use this framework for accomplishing more reasoning tasks? I hope the author can provide some analysis or experimental data to elucidate the relationship between logical reasoning and human-AI collaboration.

---

> ### Author Response · Authors · 2023-11-21
> **Rebuttal by Authors**
>
> **Q1: The experimental tasks seem somewhat weak.**
>
> A1:Thanks for your feedback. Following your suggestions, we consider incorporating additional robust human-AI collaboration tasks, including overcooked game [1], which is a benchmark environment for fully cooperative human-AI task performance. The goal of the game is to deliver soups as fast as possible. The agents should split up tasks on the fly and coordinate effectively in order to achieve high reward. The six possible actions are: up, down, left, right, noop, and "interact", which does something based on the tile the player is facing, e.g. placing an onion on a counter. Each layout has one or more onion dispensers and dish dispensers, which provide an unlimited supply of onions and dishes respectively. This game includes five layouts: Cramped Room, Asymmetric Advantages, Coordination Ring, Forced Coordination, and Counter Circuit. As good coordination between teammates is essential to achieve high returns in this environment, we use cumulative rewards over a horizon of 400 timesteps for our agents as a proxy for coordination ability. For all DRL experiments, we report average rewards across 100 rollouts and standard errors across 5 different seeds. We present quantitative results in these five layouts in the following table. There is a large gap between our method and Seq2Seq in all layouts. We will put these results into our revised manuscript.
>
> Table1: Rewards over trajectories of 400 timesteps for our methods and Seq2Seq.
> |  Layouts  | Seq2Seq | Ours |
> | :---  | ---:  | :--: |
> | Cramped Room  | 133.2(8.1) | 158.3(7.1)|
> | Asymmetric Advantage | 169.2(3.0) |185.7(5.4)|
> | Coordination Ring | 115.4(7.9) | 142.1(7.1) |
> | Forced Coordination | 71.2(4.9) | 85.6(5.7)|
> | Counter Circuit | 60.9(4.4)|89.0(3.5)|
>
> [1] Carroll M, Shah R, Ho M K, et al. On the utility of learning about humans for human-ai coordination[J]. Advances in neural information processing systems, 2019, 32.
>
> **Q2:The authors should provide the related work in the main text rather than in the appendix.**
>
> A2:Thanks for your recommendation. By highlighting the existing research and advancements in human-AI collaboration, we can demonstrate the significance of our work and the contributions we are making to this particular area. We have added related work that focuses on human-AI collaboration into our revised manuscript.
>
> **Q3: Presentation error.**
>
> A3: Thank you for your valuable suggestions. I have certainly incorporated your recommendations to improve the clarity and formatting of our paper.
>
> **Q4:Why does logical reasoning help enhance human-AI collaboration?**
>
> A4: Logical reasoning allows humans and AI systems to interpret and understand each other’s actions, intentions, and reasoning processes. By applying logical rules and principles, both parties can analyze and make sense of the information exchanged during collaboration. This shared understanding enables effective communication and coordination. It ensures coherence and consistency in the collaboration process. When humans and AI systems apply logical principles, they can identify and resolve contradictions or inconsistencies in their respective actions or decisions. This helps in aligning their efforts and maintaining a coherent and productive collaboration. By employing logical thinking, both humans and AI systems can break down complex problems into smaller components, identify patterns, and derive logical deductions to arrive at optimal solutions. We have added more analysis and reasoning experiments (e.g, overcooked games) in the supplementary material to elucidate the relationship between logical reasoning and human-AI collaboration.

---

> > ### Comment · Reviewer_wN2Q · 2023-11-22
> >
> > I thank the authors for the additional experiments and clarification.
> > I keep my score and defer to the other reviewers for the deep assessment of the significance of this work due to my limited knowledge.

---

### Author Response · Authors · 2023-11-21
**Invitation for further review feedback**

Dear Esteemed Reviewers,
We write to express our deep appreciation for the time and expertise you have devoted to reviewing our paper. Your feedback has been invaluable, and we have made every effort to address each point in our revised submission. As the review period draws to a close, we warmly invite any additional comments or suggestions you may have. Your insights are essential in refining and elevating our work.

Should you feel that our amendments have satisfactorily resolved your initial concerns, we would be grateful if you could consider reflecting this in your final assessment of our paper. Conversely, if there are any unresolved issues or further clarifications needed, please feel free to reach out. We are fully committed to engaging in further discussion and providing any additional information required.

We are thankful for your significant contributions throughout this review process and look forward to possibly engaging in more dialogue.

Thank you once again for your invaluable input.

Kind regards,

The authors

---

### Meta-Review · Area_Chair_2EbK · 2023-12-15

**Metareview:**

This paper presents a framework for human-AI collaboration using logical reasoning to develop machines that assist humans. The framework uses trajectories of human activities, logic rules, and trains agents in new environments. It includes empirical evaluations, comparisons with state-of-the-art approaches, and discussions on neuro-symbolic methods.

The reviewers generally agree on the paper's strengths, including its novel approach to integrating robots with humans, the use of logical reasoning for generalization, and strong empirical evaluations. The paper's potential contribution to the neuro-symbolic field and human-robot interaction is recognized. There are concerns about the method clarity and need for more literature comparison, but those concerns seem to be well addressed in the author responses.

Considering the strengths of the paper, the significance of its contributions, and the authors' comprehensive and satisfactory responses to the reviewers' concerns, I recommend accepting this submission and suggest revising the paper based on the reviewers' comments.

**Justification For Why Not Higher Score:**

This work is solid but might not yet have the breadth of impact that a spotlight presentation demands.

**Justification For Why Not Lower Score:**

The paper successfully addresses an important problem in human-AI collaboration, demonstrates novelty and thorough empirical evaluation, and significantly improves upon existing approaches. The authors have effectively resolved key concerns raised during the review process, strengthening the paper's overall quality and relevance.

---

### Decision · Program_Chairs · 2024-01-16

Accept (poster)